# How do project managers' competencies impact project success? A systematic literature review

**Paola Ochoa Pacheco** *, **David Coello-Montecel**, **Michelle Tello**, **Virginia Lasio**, **Alfredo Armijos**

ESPAE Graduate School of Management, Escuela Superior Politécnica del Litoral (ESPOL), Guayaquil, Ecuador

* pjochoa@espol.edu.ec

## Abstract

Despite the existence of systematic literature reviews focused on examining the factors contributing to project success, there remains a scarcity of reviews addressing the relationship between the project managers' competencies and project success. To fill this gap in the literature, this review aimed to evaluate peer-reviewed articles, published between 2010 and 2022, and analyze the impact of project managers' competencies on project success. The Web of Science, Scopus, ScienceDirect, and ProQuest electronic databases were first consulted in September 2021, with an update in August and October 2022. A total of 232 titles were analyzed. Ten articles met the criteria and were fully reviewed. A content analysis and a citation network were carried out to analyze the included articles. The analysis revealed that the existing literature has primarily explored the influence of competencies from the personal and social dimensions, such as leadership, communication, and emotional intelligence, on project success. Conversely, competencies from other dimensions have received less attention in the literature. In addition, this review contributes to the literature by providing a holistic categorization of competencies associated with project success and examining and organizing project success criteria into three dimensions.

## 1. Introduction

The profound economic, technological, and social changes experienced in recent years [1, 2] have compelled organizations to devise strategies and implement initiatives to adapt to uncertain environments [3]. Projects allow organizations to face these challenges by leveraging their expertise and capabilities to deliver solutions aligned with business objectives [4]. Project management (PM) has been acknowledged as a valuable discipline for managers and professionals implementing strategic organizational transformations [1]. Given the shortage of qualified talent to execute strategic initiatives and drive change [5], the project managers' (PMGs) competencies have garnered significant attention from scholars [6–11] and PM institutions [12, 13]. Consequently, a substantial body of literature has devoted considerable effort to delineating the competencies that have the potential to enhance projects' positive outcomes [14–18].

**Data Availability Statement:** All data are available either within the manuscript (Tables 1, 2, 3, 4) or as supplementary files. Hyperlinks are provided within the manuscript in the reference list.

**Funding:** The authors received no specific funding for this work.

**Competing interests:** The authors declare no conflict of interest.

There has been a growing interest in exploring the individual and organizational factors contributing to project success (PS). At the individual level, the PMGs' leadership style [19], job satisfaction [20], trust [21], job crafting [22], and work-family conflict [23], among other factors have been associated with PS. At the organizational level, scholars have highlighted that PS can be influenced by innovative climate [24], organizational culture [25], cultural diversity [26], governance [27], knowledge sharing and perceived trust and cohesion of the team [28, 29], among others.

Despite the existence of systematic literature reviews (SLRs) that summarize the available evidence regarding factors associated with PS [30], there remains a scarcity of SLRs focusing on PMGs' competencies [31, 32] and their impact on PS. Only a limited number of SLRs [33] have been dedicated to identifying the competencies essential for achieving PS. However, to the best of our knowledge, an SLR focused on analyzing the relationship between PMGs' competencies and PS has not been conducted before. To fill this gap in the literature, this SLR analyzes the existing evidence regarding the relationship between PMGs' competencies and PS. Therefore, the present SLR was designed to address the following research questions: (RQ1) Which PMGs' competencies are the most examined in the existing literature? (RQ2) Which success criteria are the most considered when measuring PS in the existing literature? (RQ3) Which PMGs' competencies have a relationship with PS?

This SLR contributes to the literature on the PM discipline in four ways. Firstly, it fills a gap in the existing literature by employing the SLR methodology to comprehensively synthesize the available evidence from published empirical studies concerning the relationship between PMGs' competencies and PS. Secondly, it employs a thematic analysis and a holistic perspective to categorize the PMGs' competencies associated with PS. This methodological approach provides a comprehensive framework for understanding the diverse competencies relevant to PS. Thirdly, it offers an insightful analysis of a graphical representation that showcases the primary authors and institutions that have significantly influenced the conceptualization of PMGs' competencies. Lastly, it examines the criteria utilized for measuring PS in the included articles and organizes them into three dimensions, enhancing the understanding of the multifaceted nature of PS assessment. By addressing these aspects, this SLR contributes to advancing knowledge in PM.

The subsequent sections of this paper are structured as follows. Section 2 presents the conceptualization of PMGs' competencies and PS. Section 3 outlines the procedure for conducting the SLR, encompassing the search strategy, study selection, data extraction, and analysis. The findings derived from the SLR are presented in Section 4. Lastly, the paper concludes by discussing the implications of the results, highlighting the strengths and limitations of the SLR, and offering final remarks.

## 2. Competencies and project success

This section provides an overview of the conceptualizations of competencies adopted in the PM literature, and briefly discuss the evolution of the PS dimensions.

### 2.1. Competencies

Various conceptualizations of competencies have been explored in the existing literature [16, 34–38]. Within the PM discipline, several studies [18, 39–42] have aligned with the classical definition proposed by Boyatzis [35]. According to his framework, competencies encompass the underlying characteristics of an individual, including knowledge, skills, abilities, attitudes, and more, that collectively enable the achievement of high performance. These elements have

served as a foundational basis for scholars [9, 43, 44] and institutions [12, 13], who have further expanded the scope to develop frameworks tailored explicitly to the domain of PM.

PM institutions, including the Project Management Institute (PMI) and the International Project Management Association (IPMA), have played a crucial role in the definition and development of various standards and frameworks that pertain to the competencies of PMGs [45]. Several studies [16, 46, 47] have employed these institutional standards to define competencies. The next paragraph provides a concise overview of these institutional frameworks.

According to the IPMA [13], competencies comprise the practical application of knowledges, skills, and abilities to achieve desired outcomes. This framework recognizes the interconnectedness of these elements, as proficiency entails acquiring relevant knowledge and developing skills that, when put into practice, enable professionals to manage projects effectively and successfully. Similarly, the PMI [12] defines competencies as the capability to carry out activities within a portfolio, program, or project setting that yield anticipated results based on established and accepted standards. This definition builds upon Boyatzis' [35] elements and aligns with the IPMA [13] perspective, but it also emphasizes compliance by acknowledging the significance of adhering to current regulations and guidelines to meet stakeholders' expectations. More recently, the PMI [48] introduced the concept of *power skills*, which refers to the abilities and behaviors that facilitate working with others and enable project professionals to succeed in the workplace, align projects to organizational objectives, and motivate teams to contribute value to the organization and its customers.

The scholarly literature [8, 37, 43, 44] has significantly contributed to the conceptualization of the competencies required by PMGs by incorporating key elements from the PM discipline. For instance, Hanna et al. [43] emphasized the evolving nature of projects. They argued that competencies entail the demonstrated ability to perform project activities within a dynamic environment, leading to expected outcomes based on established standards. Building upon this perspective, Bashir et al. [44] defined competencies as a meta-ability that integrates skills, aptitudes, and abilities to perform throughout the project life cycle, from initiation to closing, intending to achieve expected results. Moreover, Crawford [49] posited a close relationship between PMGs' competencies and PS. Recent literature has underscored the pivotal role of PMGs' competencies in attaining higher levels of success, enhancing efficiency and effectiveness, and consequently increasing the likelihood of PS [8].

## 2.2. Project success

This section provides an overview of the historical development of the conceptualization of PS, tracing its progression from a unidimensional to a more comprehensive and multidimensional concept [50]. It also aims to identify the dimensions and criteria incorporated into the concept in recent years. Furthermore, it defines PS and examines its distinctions from related concepts, such as project performance and efficiency.

Traditionally, scholars [39, 51–53] have viewed PS as a combination of success factors and criteria. On the one hand, success factors refer to the significant elements that enhance the probability of achieving success. On the other hand, success criteria comprise a set of measures used to evaluate if the project can be judged as successful [39]. This SLR specifically focuses on PS criteria.

The measurement criteria for assessing PS have undergone significant evolution to encompass the complex and dynamic nature of projects, resulting in the development of more comprehensive models [52, 54]. Initially, PS frameworks primarily focused on efficiency criteria, commonly referred to as the "golden triangle," "iron triangle," or "holy trinity," which encompassed elements such as time, cost, and quality [54]. Subsequent models expanded to

incorporate dimensions of client and project team satisfaction [55]. From the year 2000 onwards, the emergence of integrative models took into account additional dimensions, including realized benefits to the business or organization [56, 57], satisfaction levels of internal and external stakeholders such as end-users, suppliers, and other relevant parties [58], the impacts on the community and environment [59], long-term effects like the creation of new markets or product lines [56, 60], and investment returns [61].

The conceptual boundaries between PS, project performance, project efficiency, and PM success have often been blurred. On the one hand, PM success represents a conventional measure of PS that primarily focuses on time, cost, and quality, assessed upon project completion [62, 63]. These criteria are also called project efficiency [64]. On the other hand, project performance refers to the degree to which management practices and processes contribute to the achievement of goals and objectives, as well as the fulfillment of stakeholders' expectations. It is typically evaluated throughout project execution and upon completion [54, 65]. In contrast, PS represents a broader and multidimensional concept encompassing the achievement of goals and objectives determined by key stakeholders after project completion [63, 64], as well as the long-term impacts of the project [66].

## 3. Method

The SLR was undertaken to investigate the abovementioned research questions and followed the guidelines outlined in the Preferred Reporting Items for Systematic Reviews and Meta-Analyses (PRISMA). The protocol employed for conducting this SLR is elaborated next.

### 3.1. Search strategy

The Web of Science, Scopus, ScienceDirect, and ProQuest electronic databases were selected for this SLR. The databases were first consulted in September 2021, with an update in August and October 2022, by searching the following keywords in the title of the article: "competence," "competency," "competences," "competencies," "skill," "skills," and "project success," without any additional constraint. The search was performed by two of the authors using the following search strings:

- *Scopus database*: TITLE ((competence) OR (competency) OR (competences) OR (competencies) OR (skill) OR (skills)) AND TITLE ((project AND success))

- *Web of Science database*: TI = (competence OR competency OR competences OR competencies OR skill OR skills) AND TI = (project success)

- *ScienceDirect database*: Title: (competence OR competency OR competences OR competencies OR skill OR skills) AND (project success)

- *ProQuest database*: title((competence OR competency OR competences OR competencies OR skill OR skills)) AND title((project success))

The metadata of the records (title, authors, document type, source title, author keywords, abstract, publication year, volume number, issue number, and DOI) was exported, compared, and saved on Microsoft Excel spreadsheets to remove duplicated studies and conduct the screening process.

### 3.2. Study selection

The study selection process comprised several stages to find relevant articles for the review. The initial research resulted in 232 articles. After removing duplicated records, 172 articles

were considered for the next stages. The procedures followed by the authors are described below.

**3.2.1. Inclusion and exclusion criteria.** The inclusion and exclusion criteria for document selection in this review were based on various factors, including publication timeline, document type, language, study type, population, and context. To be included in this review, documents had to meet the following criteria: (1) they had to be peer-reviewed scholarly research articles, (2) they had to be published between January 2010 and October 2022, (3) they had to be written in English, (4) they had to have a quantitative approach measuring PMG's competencies as independent variable and PS as a dependent variable, (5) the study population had to consist of PMGs or similar positions (e.g., project director, project leader, senior PMG, department manager, functional manager, team leader), and (6) the research work had to be conducted in professional settings. The study selection process did not impose restrictions on industry, project type, or project size to ensure a broader scope and encompass various perspectives. This approach allowed for the retrieval of peer-reviewed scholarly articles that addressed the research questions of this SLR. Initially, 172 articles were evaluated, and after applying the inclusion criteria, 131 records were removed. Subsequently, 41 research articles remained for the screening process.

**3.2.2. Article screening process.** After applying inclusion and exclusion criteria, the retained articles were screened by title, abstract, and full text. This process was conducted by two of the authors independently. The reasons for excluding articles were reported in each step. The exclusion criteria were objectively applied. Studies were excluded if the relationship between PMGs' competencies and PS was not examined. Each reviewer's number and list of excluded articles were compared after the screening. In those cases where there was disagreement between reviewers, a third author reviewed the article and discussed it with the other two authors to reach a consensus. Eligible articles were included in the final review. Ineligible articles were formally excluded, with the reasons for exclusion noted.

Out of 41 articles, seven were excluded based on the title. In this step, the main reasons for exclusion were: (a) the study was related to project-based learning ($n = 4$), (b) the article was a literature review ($n = 3$), and (c) the article was a case of study ($n = 1$). The retained 34 articles were screened by abstract. After analyzing the abstract of each article, eight were removed because of the following: (a) the study had a qualitative design ($n = 1$), (b) the article was a case of study ($n = 1$), (c) the article analyzed only leadership styles ($n = 1$), (d) the article was theoretical ($n = 2$), (e) the study was not conducted in a PM professional setting ($n = 1$), and (f) the article did not analyze the relationship between PMGs' competencies and PS ($n = 2$). Finally, the full-text screening was carried out on 26 articles. Thirteen records were excluded based on the following reasons: (a) PMGs' competencies were not measured ($n = 3$), (b) the article was theoretical ($n = 3$), and (c) the study did not analyze the relationship between PMGs' competencies and PS ($n = 7$). After the whole screening process, 13 articles were considered for quality assessment.

**3.2.3. Quality assessment.** The quality assessment focused on ten quality criteria statements: (1) The research questions, objectives, or hypothesis were appropriately established; (2) The study design was well described and appropriate for answering the research questions; (3) The sample and population of the study were clearly described, and its size was sufficient to carry out the proposed analysis; (4) The response rate was reported and above 50%; (5) The instruments used for measuring PMGs' competencies were well described and design-based; (6) The instrument used for measuring PS was well described and design-based; (7) The statistical method was appropriate and sufficiently described to enable them to be repeated; (8) The research questions were adequately answered; (9) The statistical significance of associations was tested and reported; (10) The conclusions were clearly described and based on the results.

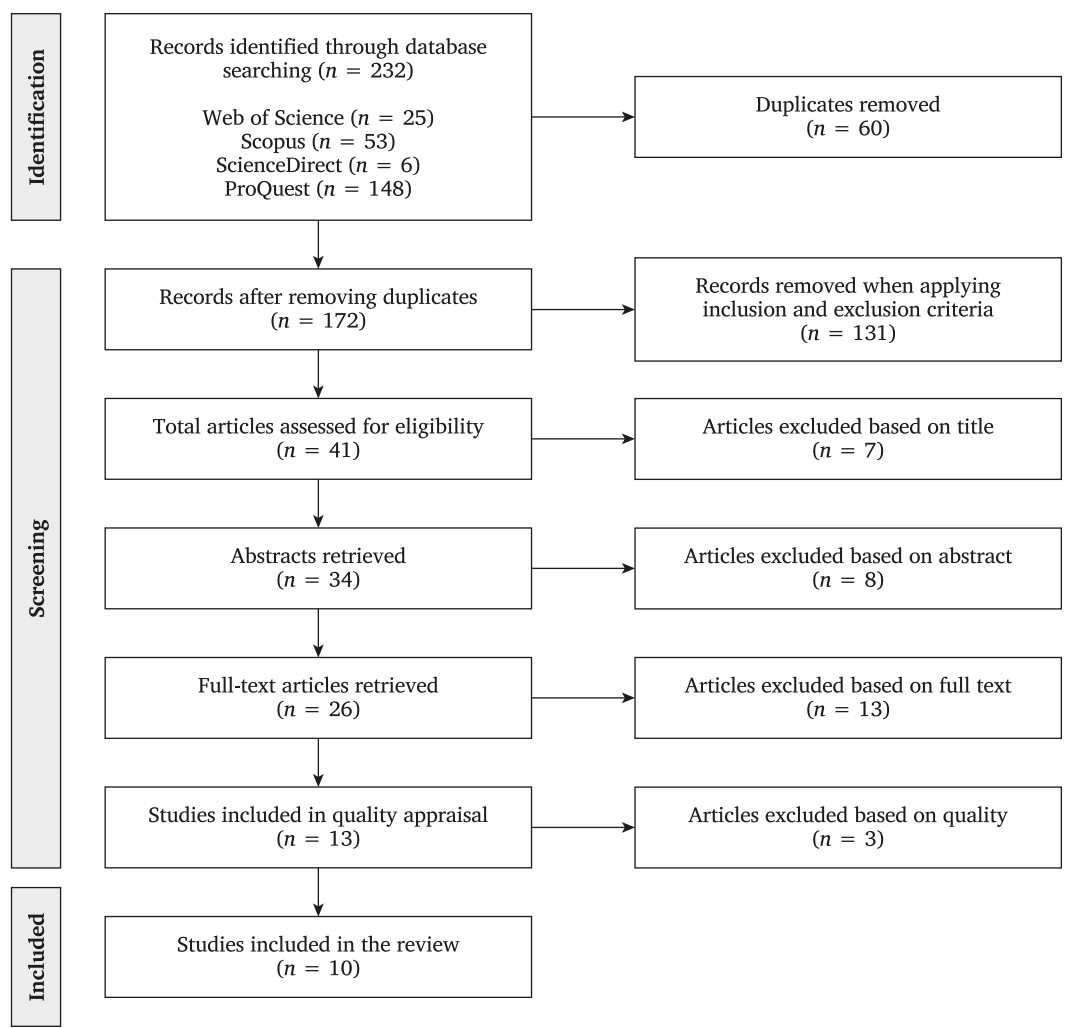

**Fig 1. PRISMA flow chart for systematic literature review.**

The abovementioned criteria were adapted from the Newcastle-Ottawa Quality Assessment Scale (adapted for cross-sectional studies), the Appraisal Tool for Cross-Sectional Studies (AXIS), and similar studies [67]. Each statement had three rating options coded as "Yes" (1 point), "No" (0 points), and "Partial" (0.5 points). Articles with a score of 7.5 points or higher were included in the final sample. The quality assessment was carried out by two authors independently. The results were compared, and the differences found were discussed to make a final decision. In this phase, three articles were removed. Ten articles were selected to conduct the analysis and answer the research questions of this SLR. Fig 1 summarizes the data extraction procedure through a PRISMA flow.

### 3.3. Data extraction and analysis

Three authors analyzed the articles for data extraction, including sample characteristics, country, setting, independent and outcome variable(s), data analysis procedures, and main findings. These data were synthesized in Table 4.

A thematic analysis was conducted to identify the dimensions of PMGs' competencies and PS criteria used in the included articles, following the procedures proposed by Nowell et al. [68]:

i. *Familiarization with the data*. The authors read and analyzed the content of each article.

ii. *Generation of initial code*s. Each author generated a list of competencies and PS criteria extracted from each article. The resulting lists were compared and matched to get a final version.

iii. *Creation of themes*. Categories were created by grouping similar competencies and PS criteria. Each of the authors carried out this process individually. The resulting lists were compared and matched to get a final version as in the previous step.

iv. *Definition and naming of themes*. Once the final list of competencies and PS were obtained and the main categories were defined, each category was named based on theoretical foundations. This process was carried out jointly by the three authors.

When studying topics such as PMGs' competencies, an important issue is how authors support their choice regarding what competencies to include in their work. This decision is important since it shapes the structure of the research field. Thus, a citation network analysis (CNA) was carried out to map the structure of the PMGs' competencies research field. In CNA, research documents serve as nodes, and the connections between them are represented by citations [69]. CNA is a practical approach for identifying contributions to a specific topic and uncovering relationships within the scholarly literature, thereby revealing patterns of influence and collaboration [70]. In this SLR, the ten included articles relied on citations of prior works to select the pertinent PMGs' competencies. These cited references were used to build a network representing the relevant frameworks in the included articles. The citation network was generated using the visNetwork package in RStudio.

## 4. Results

### 4.1. Study characteristics

The main characteristics of the articles included in this SLR are shown in Table 1. Data were collected from 11 countries across five regions: Asia, Europe, North America, Oceania, and

**Table 1. Selected publications for analysis.**

| Year | No. of published articles | Country/Region | Journal | Journal Impact Quartile | Article included in the SLR |
|---|---|---|---|---|---|
| 2022 | 2 | South America (Brazil), North America (Canada and USA), Asia (China), Europe (Ireland and Portugal) | International Journal of Managing Projects in Business | Q1 | Sampaio et al. [71] |
| | | Pakistan | Pakistan Journal of Commerce and Social Science | Q1 | Rana and Shuja [73] |
| 2021 | 3 | Egypt | Brazilian Journal of Operations and Production Management | Q2 | Elmezain et al. [74] |
| | | Pakistan | International Journal of Information Technology Project Management | Q2 | Ahmed and Lodhi [75] |
| | | Pakistan | Sustainability | Q1 | Irfan et al. [76] |
| 2020 | 2 | Brazil | International Journal of Project Organisation and Management | Q3 | Lima and Quevedo-Silva [77] |
| | | Pakistan | Pacific Business Review International | N/A | Khan et al. [78] |
| 2019 | 1 | Poland | International Journal of Managing Projects in Business | Q1 | Podgórska and Pichlak [72] |
| 2017 | 1 | Pakistan | Project Management Journal | Q1 | Maqbool et al. [15] |
| 2010 | 1 | Europe, North America, Australia, New Zealand, among others | Baltic Journal of Management | Q2 | Müller & Turner [39] |

Note: The Journal Impact Quartile was based on Scopus CiteScore. N/A = Journal Impact Quartile not available.

South America. Notably, Pakistan emerged as the most prolific country, with five papers published between 2010 and 2022, followed by the USA ($n$ = 2) and Brazil ($n$ = 2). Most studies were published within the last five years ($n$ = 9). Out of the ten articles, eight were published in journals categorized in the Q1 ($n$ = 5) and Q2 ($n$ = 3) impact quartiles. In terms of study design, most articles employed a purely quantitative approach ($n$ = 8), while two utilized mixed methods. For instance, Sampaio et al. [71] conducted a systematic review to identify the competencies to be included in their subsequent questionnaire, while Podgórska and Pichlak [72] employed a mixed-method approach comprising semi-structured interviews and a survey questionnaire.

## 4.2. Project managers' competencies

### 4.2.1. The most influential theoretical frameworks.
The majority of articles ($n$ = 9) included in the SLR employed an existing framework to identify the PMGs' competencies that were examined in their empirical analyses. However, in the study conducted by Sampaio et al. [71], a comprehensive literature review was undertaken to determine the specific competencies that should be considered for testing their impact on PS.

A CNA was conducted to explore the interrelationships among the ten articles included in this SLR and to identify the most influential frameworks for defining and determining the PMGs' competencies. Fig 2 visually represents the articles included in the SLR as square nodes

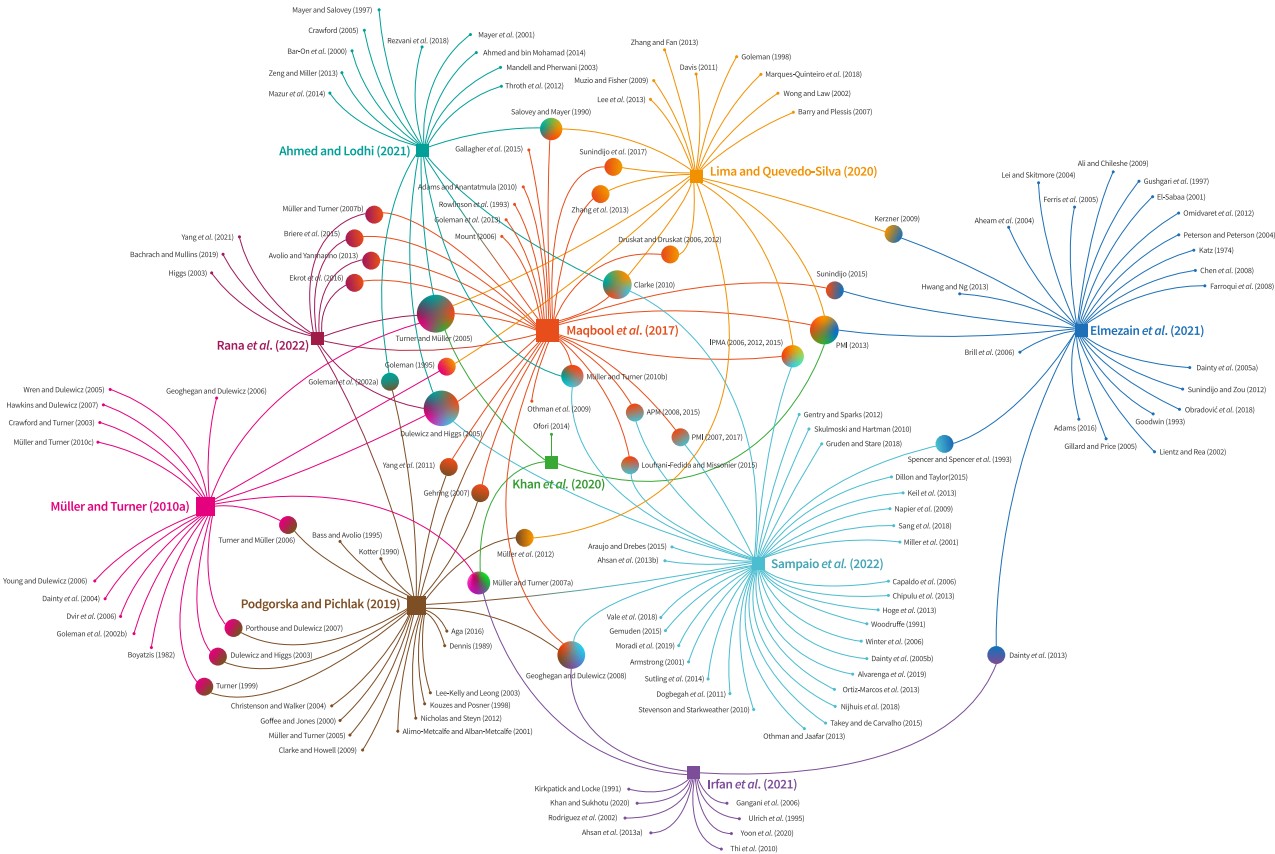

**Fig 2. The PMGs' competencies research network from a citation network perspective.** Notes. Square nodes represent the articles in the SLR ($n$ = 10), while circle nodes denote the studies that contributed to conceptualizing PMGs' competencies. The number of citations gives the size of the node.

and the studies that have contributed to conceptualizing PMGs' competencies as circle nodes. The size of each node reflects the number of citations it has received. The diagram layout was arranged such that the most frequently cited studies are positioned in the center, while less frequently cited ones are placed towards the periphery. A summary of the most influential works in PMGs' competencies is provided below.

Four articles included in this SLR [39, 72, 73, 75] employed a common framework developed by Dulewicz and Higgs [79]. This framework encompasses 15 leadership competencies categorized into three dimensions: intellectual competencies (critical analysis and judgment, vision and imagination, strategic perspective), managerial competencies (managing resources, engaging communication, empowering, developing, achieving), and emotional competencies (self-awareness, emotional resilience, intuitiveness, interpersonal sensitivity, influence, motivation, conscientiousness). Additionally, two articles [15, 77] drew upon Clarke's [80] study, which identified four main PMGs' competencies: communication, teamwork, attentiveness, and managing conflict. Other frameworks utilized in the SLR articles were proposed by Sunindijo [81], Katz [82], Nguyen and Hadikusumo [83], and Ofori [84]. These frameworks shared common elements, emphasizing the significance of communication, leadership, managing emotions, and interpersonal relationships as essential competencies for PMGs. Notably, the Project Manager Competency Development Framework [12] and the Individual Competence Baseline for Project Management [13] were among the most cited institutional frameworks employed in the SLR articles.

**4.2.2. Categorization of project managers' competencies.** Several common competencies were identified based on the review of competencies reported in each article. These competencies were categorized into four dimensions based in previous studies [11, 31, 85], as presented in Table 2: cognitive, personal, social, and sustainability. It should be noted that not all competencies were consistently referred to by the same name across the included articles. Therefore, the names used to denote a specific competence in each article are listed in the third column of Table 2.

## 4.3. Project success criteria

Previous literature has traditionally focused on PS measures related to cost, time, and quality. However, the findings of the SLR indicate a growing tendency to incorporate a broader range of success criteria. Table 3 presents a categorization of the different success criteria reported in the included articles.

The first dimension pertains to the impact on stakeholders, encompassing clients, users, providers, the project team, and other relevant parties. While stakeholder impact is commonly assessed through satisfaction measures, some studies consider alternative indicators such as the acceptability of the product, perceived benefits [73], or the fulfillment of stakeholder expectations [74]. Less frequently addressed are measures related to the impact on the organization, for which two criteria were identified: (i) visible short-term improvements in organizational outcomes or performance [73, 75, 76], and (ii) long-term improvements, such as the development of new technologies or the initiation of future projects [39, 72, 75]. Additional criteria related to the project management process were identified, encompassing project performance, achievement of the project's primary objectives, other self-defined criteria related to project management, and compliance with procedures, safety regulations, and environmental standards. Project performance indicators include the traditional metrics of cost, time, and quality of the project's deliverables [39, 72].

**Table 2. PMGs' competencies in the articles included in the SLR (*n* = 10).**

| Dimensions | Competencies | Associated terms | Authors | | | | | | | | | |
|---|---|---|---|---|---|---|---|---|---|---|---|---|
| | | | Ahmed & Lodhi [75] | Elmezain et al. [74] | Irfan et al. [76] | Khan et al. [78] | Lima & Quevedo-Silva [77] | Maqbool et al. [15] | Müller and Turner [39] | Podgórska and Pichlak [72] | Rana and Shuja [73] | Sampaio et al. [71] |
| Cognitive | Creativity | Creativity, vision and imagination, resourcefulness, creative thinking, imagination | | | | | | | X | X | X | X |
| | Decision-making | Decision-making skills, intuitiveness, critical analysis, and judgment | | | X | | | | X | X | X | |
| | Strategic perspective | Strategic perspective | | | | | | | X | X | X | X |
| Personal | Conscientiousness | Conscientiousness | | | | | | | X | X | X | |
| | Emotional intelligence | Emotional intelligence, self-awareness, resilience, stress management, self-control, sensitivity | X | X | | X | X | X | X | X | X | X |
| | Results orientation | Results orientation, motivation, achieving | | | | | | | X | X | X | X |
| Social | Communication | Engaging communication, communication management, communication, verbal skills, feedback | | | | X | X | X | X | X | X | X |
| | Conflict management | Managing conflict, resolving conflicts | | | X | | X | X | | | | |
| | Interpersonal relations | Attentiveness, interpersonal linkages, interpersonal skills, influence | | X | X | X | X | | X | X | | |
| | Leadership | Political skills, transformational leadership, empowering, developing, coaching | | X | X | | | X | X | X | X | X |
| | Teamwork | Team building, teamwork | | | X | | X | X | | | | |
| Sustainability | Ethics | Transparency, honesty, integrity | | X | | | | | | | | X |

## 4.4. Empirical analysis of the relationship between project managers' competencies and project success

Table 4 presents a comprehensive overview of the research methods and results employed in the included studies. Several studies conducted correlational analyses to examine the relationship between competencies and various PS criteria [71, 72, 75], as well as overall PS [15].

**Table 3. Project criteria success in included articles.**

| Project success dimensions | Success criteria | Articles included in the SLR | | | | | | | | |
|---|---|---|---|---|---|---|---|---|---|---|
| | | Ahmed & Lodhi [75] | Elmezain et al. [74] | Irfan et al. [76] | Lima & Quevedo-Silva [77] | Maqbool et al. [15] | Müller and Turner [39] | Podgórska and Pichlak [72] | Rana and Shuja [73] | Sampaio et al. [71] |
| Impact on stakeholders | Clients satisfaction | X | X | | | | X | X | X | |
| | Users satisfaction | | | X | | X | X | X | X | X |
| | Providers satisfaction | | | | | X | X | X | | |
| | Teams satisfaction | X | | | | X | X | X | X | |
| | Other stakeholders satisfaction | | | | | X | X | X | | |
| Impact on organization | Visible short-term improvements | X | | X | | | | | X | |
| | Long-term improvements | X | | | | | X | X | | |
| General project management | Project efficiency (time, cost, and quality) | X | X | X | X | X | X | X | X | X |
| | Achieving purpose and objectives | | | X | X | X | X | X | X | X |
| | Project self-defined criteria | | | | | X | X | X | | |
| | Compliance with safety and environmental procedures and regulations | | | X | | | | | | |

Regression analysis was a common method to assess the predictive impact of PMGs' competencies on PS criteria in the selected articles [39, 72, 74, 75]. Additionally, some studies [73, 74, 76, 78] employed structural equation modeling (SEM) or partial least squares (PLS) to analyze the predictive effect of PMGs' competencies, modeled as second-order constructs, on PS.

**4.4.1. Relationship between cognitive competencies and project success.** Cognitive competencies encompassed creativity, decision-making, and strategic perspective. Findings from two studies revealed a positive correlation between creativity and various PS criteria, such as accomplishing project objectives, project efficiency, user satisfaction [71], and suppliers' satisfaction [72]. While creativity significantly predicted project efficiency, its effect on achieving project objectives and user satisfaction was not statistically significant [71]. Two studies included in the SLR [39, 72] provided evidence concerning the relationship between strategic perspective and PS. Firstly, Müller and Turner [39] found that this competence influences project efficiency and self-defined success criteria. Secondly, Podgórska & Pichlak [72] reported that strategic perspective is significantly associated with all the analyzed PS criteria except for project efficiency and self-defined success criteria. Regarding the decision-making competence, Müller and Turner [39] did not identify any significant predictive effects of this competence. However, Podgórska and Pichlak [72] observed significant positive correlations between decision-making and all the PS criteria, with the highest coefficients observed for self-defined success criteria, end-user satisfaction, and satisfaction of other stakeholders.

**4.4.2. Relationship between personal competencies and project success.** Personal competencies included emotional intelligence, results orientation, and conscientiousness. Among these competencies, emotional intelligence has received significant attention in the included studies. Out of the ten studies, eight explored the relationship between emotional intelligence and PS. The evidence revealed direct and significant predictive effects of emotional intelligence on various PS criteria, such as end-user satisfaction [71], achievement of project objectives [39], and overall PS [15].

**Table 4. Summary of included articles.**

| No. | Author | Sample | Sector | Response rate | Analysis tool | Main results |
|---|---|---|---|---|---|---|
| 1 | Sampaio et al. [71] | *Sample size*: N = 121 *Age*: 20–40 years: ~64% > 40 years: ~36% *Gender*: Male: ~74% Female: ~26% *Working experience*: < 11 years: ~25% 11–20 years: ~48% > 20 years: ~27% | Information systems | 51% | Correlation analysis PLS-SEM | • Project performance was significantly associated with leadership, results orientation, emotional intelligence, ethics, creativity, and motivation. • Goal achievement was significantly associated with leadership, emotional intelligence, ethics, and creativity. • User satisfaction was significantly associated with leadership, communication, results orientation, ethics, creativity, and motivation. • Emotional intelligence was a significant predictor of user satisfaction. • Ethics was a significant predictor for goal achievement. • Creativity and motivation were significant predictors of project performance. |
| 2 | Rana and Shuja [73] | *Sample size*: N = 183 *Age*: 21–30 years: ~52% 31–40 years: ~36% > 40 years: ~12% *Gender*: Not reported *Working experience*: <6: ~48% > 5 years: ~52% | Transportation | NR | SEM | • Leadership competencies, measure as a second-order construct, was not a significant predictor of PS. • Innovative work behavior fully mediates the relationship between leadership competencies and PS. |
| 3 | Elmezain et al. [74] | *Sample size*: N = 400 *Age*: < 30 years: ~13% 30–49 years: ~52% > 50 years: ~35% *Gender*: Male: ~ 78% Female: ~ 22% *Experience*: < 7 years: ~29% 8–20 years: ~58% >20 years: ~13% | Construction | 87% | Regression analysis | • Human and political skills were a significant predictor of PS. |
| 4 | Ahmed & Lodhi [75] | *Sample size*: N = 112 *Age*: Not reported *Gender*: Male: ~ 79% Female: ~ 21% *Experience*: < 5 years: ~17% 5–10 years: ~59% >15 years: ~23% | Information technology and telecommunication public projects | 75% | Correlation analysis Regression analysis | • Self-awareness and resilience were significantly associated with the following PS criteria: organizational success, team satisfaction, user satisfaction, and project performance. • Emotional intelligence was a significant predictor of all PS criteria. |

*(Continued)*

**Table 4.** (Continued)

| No. | Author | Sample | Sector | Response rate | Analysis tool | Main results |
|---|---|---|---|---|---|---|
| 5 | Irfan et al. [76] | *Sample size*: N = 260 *Age*: Not reported *Gender*: Male: ~ 78% Female: ~ 22% *Experience*: 5–15 years: ~73% > 15 years: ~27% | Public sector organizations | 52% | PLS-SEM | • PMGs' competencies, measured as a second-order construct, were not a significant predictor of PS. |
| 6 | Lima and Quevedo-Silva [77] | *Sample size*: N = 119 *Age*: mean = 40 years *Gender*: Male: ~ 81,51% Female: ~ 18,49% *Working experience*: mean = 10 years | Not reported | 88% | PLS-SEM | • Interpersonal skills were a significant predictor of PS. • Emotional intelligence was not a significant predictor of PS. |
| 7 | Khan et al. [78] | *Sample size*: N = 255 *Age*: 25–45 years: ~63% 46–55 years: ~33% >55 years: ~4% *Gender*: Male: ~ 100% *Experience*: 1–5 years: ~11% 6–10 years: ~55% 11–15 years: ~34% | Construction | NR | PLS-SEM | • Communication and interpersonal skills were a significant predictor for PS. |
| 8 | Podgórska and Pichlak [72] | *Sample size*: N = 102 *Age*: >41 years: ~ 63% > 40 years: ~37% *Gender*: Male: ~ 31% Female: ~ 69% *Working experience*: < 6 years: ~64% > 5 years: ~36% | Not reported | 58% | Regression analysis | • Conscientiousness was a significant predictor of PS in high-complexity and mandatory projects. • Communication was a significant predictor of PS in medium-complexity projects. • Motivation was a significant predictor of PS in repositioning projects. |

(*Continued*)

**Table 4.** (Continued)

| No. | Author | Sample | Sector | Response rate | Analysis tool | Main results |
|---|---|---|---|---|---|---|
| 9 | Maqbool et al. [15] | *Sample size*: N = 345 *Age*: 25–45 years: ~ 65% > 45 years: ~35% *Gender*: Not reported Working experience: 5 to 15 years: ~67% > 15 years: ~33% | Construction | 84% | Correlation analysis Regression analysis | • Self-awareness and relationship management were significantly associated with PS. <br> • Communication reported the highest correlation with PS among all competencies included in the study. <br> • PMGs' competencies, measured as a second-order construct composed by communication, teamwork, attentiveness and managing conflict, were significantly associated with PS. <br> • Emotional intelligence was a significant predictor of PS. |
| 10 | Müller and Turner [39] | *Sample size*: N = 400 *Age*: 40-year-old or younger: ~ 27% > 40 years: ~73% *Gender*: Male: ~ 65% Female: ~ 34% *Working experience*: Not reported | Not reported | Not applicable (snowball sampling) | Regression analysis | • Strategic perspective was a significant predictor of the following PS criteria: project performance and self-defined criteria of success. <br> • Managing resources perspective was a significant predictor for user satisfaction, other stakeholders' satisfaction, meeting user requirements, customer satisfaction and reoccurring business. <br> • Empowering was a significant predictor for team satisfaction. <br> • Self-awareness was a significant predictor for project achievement. <br> • Interpersonal sensitivity was a significant predictor for supplier satisfaction. <br> • Influence was a significant predictor for other stakeholder satisfaction. <br> • Motivation was a significant predictor for project performance. <br> • Conscientiousness was a significant predictor for team satisfaction. |

Notes. PLS-SEM: Partial Least Squares Structural Equation Modeling, SEM: Structural equation modeling, PS: Project success, PMGs: Project managers, NR: Non-reported.

Regarding results orientation, Sampaio et al. [71] demonstrated that this competence had a predictive effect on project efficiency. Correlational analysis revealed a higher correlation between this competence with PS criteria related to user satisfaction [71] and satisfaction of other stakeholders [72]. The conscientiousness competence was examined in the studies conducted by Müller and Turner [39] and Podgórska and Pichlak [72]. This competence emerged as a significant predictor of team satisfaction and the achievement of project objectives [39]. Furthermore, the effects of conscientiousness could vary depending on the type and complexity of the project [72].

**4.4.3. Relationship between social competencies and project success.** Based on the articles included in this review, social competencies, such as communication, leadership, interpersonal relations, conflict management, and teamwork, tend to be associated with PS. Leadership has been extensively studied in the project management literature and was addressed in seven out of the ten articles included in this review. Correlational analysis revealed that leadership shows significant associations with nearly all PS criteria, being its highest correlation with the user satisfaction criterion [71, 72]. Regarding its predictive effect on PS, Sampaio et al. [71] reported a non-significant effect of this competence on some

criteria, such as achieving project's purpose, project efficiency, and stakeholders' satisfaction, while other studies found a significant effect on team satisfaction criterion [39] and an overall PS measure [15, 74].

Regarding communication, correlational analysis showed that this competence is highly correlated with stakeholders' satisfaction criterion [71, 72]. Maqbool et al. [15] found that communication had the strongest correlation with a general measure of PS among all competencies included in their study. The predictive effect of this competence on PS was confirmed by Lima and Quevedo-Silva [77], Khan et al. [78] and Podgórska and Pichlak [72]. However, Sampaio et al. [71] and Müller and Turner [39] reported non-significant effects of communication of PS criteria.

Interpersonal relations showed significant positive associations with PS criteria, with the strongest coefficients on achieving project's purpose [72]. Müller and Turner [39] found that this competence has a significant predictive effect on other stakeholders' satisfaction criteria, while two studies [77, 78] reported its predictive effect on a general PS measure. Finally, significant positive associations of conflict management and teamwork with PS were reported by Maqbool et al. [15]. However, its predictive effect on individual PS criteria were not estimated on any of the included articles.

**4.4.4. Relationship between sustainability competencies and project success.** According to Elmezain et al. [74], the capacity to demonstrate integrity, sincerity, and authenticity, and to inspire confidence and trust in others, is relevant for achieving PS. The authors emphasized that PMGs who possess integrity play a crucial role in the advancement of any project. Similarly, Sampaio et al. [71] highlighted ethics, conceptualized as transparency, integrity, and honesty, as the most significant competence for achieving PS in terms of goal attainment.

# 5. Discussion

This SLR examined the evidence pertaining to the relationship between PMGs' competencies and PS. The analysis of the included studies yielded three key findings. Firstly, six distinct clusters of authors were identified, each contributing to the conceptualization and identification of PMGs' competencies. Secondly, the conceptualization of PS has evolved from a traditional approach centered around criteria such as time, cost, and quality, to a more comprehensive, holistic, and multidimensional perspective. Lastly, through thematic analysis, a total of 12 competencies, organized into four dimensions, were identified as potential determinants of PS. Notably, the most significant competencies associated with PS were found within the personal and social dimensions. A brief discussion of these findings is presented below.

In relation to the first finding, this SLR identified six distinct clusters of authors whose work influenced the competence frameworks utilized in the included articles. These clusters represented conceptualizations proposed by scholars and reputable PM institutions. The in-depth content analysis revealed that the frameworks proposed by Dulewicz and Higgs [79] and Clarke [80] were the most prevalent among the examined articles. Conversely, frameworks developed by Sunindijo [81], Ofori [84], and Nguyen and Hadikusumo [83] were comparatively less frequently employed. Additionally, the PMI [12] emerged as a key institutional point of reference for identifying the competencies required in a PMG. For instance, Lima and Quevedo-Silva [77] and Maqbool et al. [15] studies adopted Clarke's [80] framework, which was based on the PMI's [12] (2017b) list of competencies. Elmezain et al. [74], who cited Sunindijo et al. [81] as their framework source, incorporated several competencies defined by the PMI [5], although the majority of these were technical.

Regarding the second finding, the articles examined in this SLR provided support for the view that PS should be understood as a multidimensional construct. This finding aligns with a

recent study by Ika and Pinto [54] that revisited the conceptualization of PS. The results of this review indicate that project performance, encompassing time, cost, and quality, emerged as the most commonly considered criterion of success across all the articles. However, a significant number of the included articles also acknowledged additional criteria, leading to the identification of three dimensions of PS. The first dimension refers to the impact on stakeholders and includes criteria related to the satisfaction of various project stakeholders, including clients, users, suppliers, and the project team, among others. The second dimension focuses on the impact of the project on the organization, comprising both short- and long-term improvements. Lastly, the third dimension is related with the general management of the project. This dimension encompasses aspects such as project performance, which includes the traditional "iron triangle" of time, cost, and quality, as well as the achievement of project objectives, adherence to project-defined criteria, and compliance with safety and environmental protocols and regulations. This conceptualization supports the multidimensional nature of PS. However, as noted by Ika and Pinto [54], it is important to highlight that the majority of the included articles overlooked the inclusion of sustainability criteria. Among the entire sample of studies, only one [76] out of ten explicitly addressed compliance with safety and environmental regulations as a criterion of success.

The findings of this SLR have provided insights into the competencies that exhibit a significant relationship with PS. Specifically, the articles included in this review extensively examined competencies associated with the personal and social dimensions, such as leadership, communication, and emotional intelligence. These competencies have been extensively studied in previous literature [18, 86], and their impact on PS was explored in the majority of the reviewed articles. Conversely, the influence of other competencies, such as ethics, received less attention and was not extensively explored. Moreover, the empirical evidence gathered in this review suggests that the effect of project management competencies on PS may vary depending on several factors. For instance, the type of project was found to be a significant factor influencing the relationship between competencies and PS [72]. Furthermore, individual and organizational factors were identified as potential mediating variables that could affect the relationship between competencies and PS [73]. These findings highlight the complexity and contextual nature of the relationship between competencies and PS. Next, a brief discussion will be presented to shed light on how these identified competencies can contribute to enhancing PS.

Leadership competence was one of the most studied competencies that improve PS. Although a few of the studies included in this SLR [71, 73] reported that it does not have a significant effect on PS, a great number of the studies [15, 39, 72, 74, 76] suggested that PMGs' leadership, conceived as their capacity to influence, empower and develop others, has a positive effect on PS. This finding agrees with the existing literature that has examined its influence on PS [87–90]. The development of competencies such as leadership allows PMGs to motivate their teams to be more productive [91], to show outstanding performance beyond expectations [89], to enhance team cohesion and engagement [92], to foster knowledge transfer across project teams [89], among other positive behaviors that would impact on projects' outcomes.

The articles included in this SLR demonstrate a significant and positive relationship between communication and PS [15, 72, 77, 78], in agreement with previous research findings [93, 94]. The significance of this competence lies in its impact throughout various stages of a project [94]. Effective communication between PMGs and the project team's members allows better collaboration [95], encourages knowledge sharing [96], and enhances the team's motivation a sense of inclusivity [94], which contribute to the overall achievement of PS.

The influence of emotional intelligence on PS was assessed in most of the articles included in this SLR. Although some studies reported a non-significant relationship between emotional intelligence [71, 73, 77], there was evidence supporting a positive association between these

two variables [15, 39, 72, 75]. PMGs with high emotional intelligence are more likely to establish stronger relationships with their teams, thereby improving communication, clarity of mission, and support, ultimately enhancing PS [21]. In addition, the development of this competence allows PMGs to adequately regulate their emotions in complex situations, promoting positive behaviors such as empathy, respect, and leadership. These behaviors contribute to their ability to address challenges successfully and ensure higher PS [97, 98].

Regarding the influence of PMGs' ethics, a positive relationship was identified between this competence and PS criteria, particularly goal achievement [71]. Ethics has been acknowledged as a driving force for the advancement of the PM profession [48] and an essential competence that PMGs should possess [99, 100]. However, empirical evidence on the impact of ethics on PS remains limited. Some related terms, such as honesty, integrity, and transparency [71, 74], or ethical thinking [100], ethical decision-making, and ethics sensitivity have been addressed in previous studies. However, its effect on PS has rarely been estimated and reported. The evidence found on ethics in this SLR was obtained from information systems and construction projects. Future studies could explore the influence of this competence in different industries and countries.

## 6. Limitations and strengths

While this review contributes with some insights to the PM literature, it is important to mention its limitations. Firstly, the time frame of the review from 2010 to 2022 may have resulted in the exclusion of relevant articles that explore the relationship between PMGs' competencies and PS. It is possible that some studies conducted outside this timeframe may provide further insights into the topic. Secondly, the use of specific search terms such as "competence," "competency," "competences," "competencies," "skill," and "skills" may have excluded other studies [86, 88] that examined the impact of different competencies individually. Including an exhaustive list of competencies in the search strings could have introduced significant heterogeneity into the reviewed articles, potentially limiting the ability to provide a comprehensive review of the existing literature.

Despite these limitations, this SLR makes several notable contributions to the PM discipline. First, it fills a gap in the existing literature by synthesizing available empirical evidence on the relationship between PMGs' competencies and PS. Second, the review conducts a thematic analysis and adopts a holistic perspective to categorize the PMGs' competencies that are associated with PS. Third, this review highlights the primary authors and PM institutions that have significantly influenced the conceptualization of PMGs' competencies. Four, the review examines the criteria used to measure PS in the included articles and organizes them into three dimensions, offering a nuanced understanding of the multifaceted nature of PS measurement.

## 7. Conclusions

The present SLR extends the literature in project management concerning the influence of PMGs' competencies on PS. Despite the growing interest in addressing the role of PMGs' competencies to achieve higher success, to the best of our knowledge, there is a lack of systematic reviews that present an analysis of the available evidence on the relationship between PMGs' competencies on PS. To fill this gap in the literature, this SLR analyzed the existing evidence regarding this relationship. Three main conclusions can be derived from the findings of this review. First, the existing literature has primarily explored the influence on PS of PMGs' competencies from the personal and social dimensions, such as leadership, communication, and emotional intelligence. Second, PS is a multidimensional construct that comprises three main

dimensions: impact on stakeholders, impact on the organization, and general project management. Third, the available data suggested that greater levels of PMGs' competencies are associated with improved PS. These findings may support scholars and managers to understand the mechanisms through which individual characteristics, such as competencies, may allow PMGs to achieve better outcomes.

This SLR contributes to the existing literature in the PM discipline by offering a comprehensive synthesis of empirical evidence, providing a thorough overview of the current state of knowledge regarding the relationship between PMGs' competencies and PS. In addition, this SLR identifies key contributors and sources of knowledge in the field, offering a valuable reference point for further research and exploration. The study also offers a review on how PS is conceptualized and measured. Moreover, it presents a classification of PMGs' competencies that influence PS. Through a thematic analysis of the competencies examined in the included articles, this categorization provides valuable insights into the emphasis placed on different types of competencies. It highlights the significant attention given to personal and social competencies, while pointing out the relatively limited exploration of sustainability, cultural, or digital competencies [85].

## Supporting information

**S1 Checklist. PRISMA 2020 checklist.**
(PDF)

**S1 Table. Inclusion and exclusion criteria used in the SLR.** Notes: PMG = Project manager, PS = Project success.
(PDF)

**S2 Table. Quality assessment criteria scoring guide.** Notes: PMG = Project manager, PS = Project success.
(PDF)

**S3 Table. Quality assessment results.** Notes: QC1 = Research questions; QC2 = Study design; QC3 = Sample representativeness; QC4 = Response rate; QC5 = PMG's competencies measurement; QC6 = PS measurement; QC7 = Statistical analysis; QC8 = Results; QC9 = Statistical significance; QC10 = Conclusions; SLR = Systematic literature review.
(PDF)

**S4 Table. Brief description of the Project Managers' competencies in included articles.**
(PDF)

**S5 Table. Brief description of the project success criteria in included articles.**
(PDF)

## Author Contributions

**Conceptualization:** Paola Ochoa Pacheco, Virginia Lasio.

**Formal analysis:** David Coello-Montecel, Virginia Lasio.

**Investigation:** Michelle Tello.

**Methodology:** David Coello-Montecel, Michelle Tello.

**Software:** Michelle Tello.

**Supervision:** Paola Ochoa Pacheco.

**Writing – original draft:** David Coello-Montecel, Michelle Tello.

**Writing – review & editing:** Paola Ochoa Pacheco, David Coello-Montecel, Virginia Lasio, Alfredo Armijos.

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
