## [Decision Letter · Decision Letter 0]

18 Sep 2023

PONE-D-23-22151How do project managers’ competencies impact project success? A systematic literature reviewPLOS ONE

Dear Dr. Ochoa,

Thank you for submitting your manuscript to PLOS ONE. After careful consideration, we feel that it has merit but does not fully meet PLOS ONE’s publication criteria as it currently stands. Therefore, we invite you to submit a revised version of the manuscript that addresses the points raised during the review process.

We look forward to receiving your revised manuscript.

Kind regards,

Jamshid Ali

Academic Editor

PLOS ONE

Additional Editor Comments:

Dear Author(s)

Your manuscript has been reviewed and you are advised to implement all the comments with true spirits.

Reviewers' comments:

Reviewer's Responses to Questions

**Comments to the Author**

1. Is the manuscript technically sound, and do the data support the conclusions?

Reviewer #1: Yes

Reviewer #2: Partly

2. Has the statistical analysis been performed appropriately and rigorously? 

Reviewer #1: Yes

Reviewer #2: Yes

3. Have the authors made all data underlying the findings in their manuscript fully available?

Reviewer #1: Yes

Reviewer #2: Yes

4. Is the manuscript presented in an intelligible fashion and written in standard English?

Reviewer #1: Yes

Reviewer #2: Yes

5. Review Comments to the Author

Reviewer #1: Your research is remarkably comprehensive and demonstrates a significant depth of inquiry into the subject matter. The thoroughness of your investigation is commendable and greatly enriches the scholarly discourse in this field.

While your detailed methodology in the introduction is informative, it might benefit from a more concise presentation. Nevertheless, the context is well-established, providing a clear understanding of your research objectives and approach.

I would also suggest considering the incorporation of psychological references where applicable. This could offer valuable insights into the psychological aspects of your study, further strengthening the theoretical foundation and enhancing the interdisciplinary nature of your work.

Overall, your research is a noteworthy contribution to the field, showcasing a commendable level of rigor and dedication. With minor adjustments to the introductory methodology and the inclusion of relevant psychological references, your paper has the potential to resonate even more deeply with both specialists and a broader readership interested in the subject matter. Keep up the excellent work!

Reviewer #2: All suggestions are mentioned in the Word file attached. The authors should check the "descriptions" in the Word file. After revision is made, I want to see the paper. If the authors make all revisions, I will accept this paper.

6. PLOS authors have the option to publish the peer review history of their article (what does this mean?). If published, this will include your full peer review and any attached files.

Reviewer #1: No

Reviewer #2: **Yes: **Teoman Erdağ

---

## [Author Response · Author response to Decision Letter 0]

10 Oct 2023

We thank your comments and suggestions for improving our articles. A summary of the changes implemented in the manuscript is presented in a letter for each reviewer. These letters were attached during the submission process.

---

## [Decision Letter · Decision Letter 1]

21 Nov 2023

How do project managers’ competencies impact project success? A systematic literature review

PONE-D-23-22151R1

Dear Dr. Ochoa,

We’re pleased to inform you that your manuscript has been judged scientifically suitable for publication and will be formally accepted for publication once it meets all outstanding technical requirements.

Kind regards,

Jamshid Ali

Academic Editor

PLOS ONE

Additional Editor Comments (optional):

Reviewers' comments:

Reviewer's Responses to Questions

**Comments to the Author**

1. If the authors have adequately addressed your comments raised in a previous round of review and you feel that this manuscript is now acceptable for publication, you may indicate that here to bypass the “Comments to the Author” section, enter your conflict of interest statement in the “Confidential to Editor” section, and submit your "Accept" recommendation.

Reviewer #2: All comments have been addressed

Reviewer #3: All comments have been addressed

2. Is the manuscript technically sound, and do the data support the conclusions?

Reviewer #2: Yes

Reviewer #3: Yes

3. Has the statistical analysis been performed appropriately and rigorously? 

Reviewer #2: Yes

Reviewer #3: Yes

4. Have the authors made all data underlying the findings in their manuscript fully available?

Reviewer #2: Yes

Reviewer #3: Yes

5. Is the manuscript presented in an intelligible fashion and written in standard English?

Reviewer #2: Yes

Reviewer #3: Yes

6. Review Comments to the Author

Reviewer #2: All is done. I belive that your paper will be helpfull to literature and sector. I accept it. Thank you.

Reviewer #3: (No Response)

7. PLOS authors have the option to publish the peer review history of their article (what does this mean?). If published, this will include your full peer review and any attached files.

Reviewer #2: **Yes: **Teoman Erdağ

Reviewer #3: **Yes: **Sharmin Akhtar

---

## [Editor Report · Acceptance letter]

28 Nov 2023

PONE-D-23-22151R1 

How do project managers’ competencies impact project success? A systematic literature review 

Dear Dr. Ochoa:

I'm pleased to inform you that your manuscript has been deemed suitable for publication in PLOS ONE. Congratulations! Your manuscript is now with our production department. 

Kind regards, 

on behalf of

Dr. Jamshid Ali 

Academic Editor

PLOS ONE